

# Genetic analysis of Thai cattle reveals a Southeast Asian indicine ancestry

Pongsakorn Wangkumhang[1], Alisa Wilantho[1], Philip J. Shaw[2],
Laurence Flori[3,4], Katayoun Moazami-Goudarzi[4], Mathieu Gautier[5],
Monchai Duangjinda[6], Anunchai Assawamakin[7] and Sissades Tongsima[1]

[1] Genome Technology Research Unit, National Center for Genetic Engineering and Biotechnology, Pathum Thani, Thailand
[2] Medical Molecular Biology Research Unit, National Center for Genetic Engineering and Biotechnology, Pathum Thani, Thailand
[3] UMR INTERTRYP, CIRAD, Montpellier, France
[4] UMR 1313 GABI, INRA, Jouy-en-Josas, France
[5] UMR CBGP (INRA/CIRAD/IRD/Supagro), INRA, Montferrier-sur-Lez, France
[6] Department of Animal Science, Faculty of Agriculture, Khon Kaen University, Khon Kaen, Thailand
[7] Department of Pharmacology, Faculty of Pharmacy, Mahidol University, Bangkok, Thailand

Corresponding author
Sissades Tongsima,
sissades@biotec.or.th

## ABSTRACT

Cattle commonly raised in Thailand have characteristics of *Bos indicus* (zebu). We do not know when or how cattle domestication in Thailand occurred, and so questions remain regarding their origins and relationships to other breeds. We obtained genome-wide SNP genotypic data of 28 bovine individuals sampled from four regions: North (Kho-Khaolampoon), Northeast (Kho-Isaan), Central (Kho-Lan) and South (Kho-Chon) Thailand. These regional varieties have distinctive traits suggestive of breed-like genetic variations. From these data, we confirmed that all four Thai varieties are *Bos indicus* and that they are distinct from other indicine breeds. Among these Thai cattle, a distinctive ancestry pattern is apparent, which is the purest within Kho-Chon individuals. This ancestral component is only present outside of Thailand among other indicine breeds in Southeast Asia. From this pattern, we conclude that a unique *Bos indicus* ancestor originated in Southeast Asia, and native Kho-Chon Thai cattle retain the signal of this ancestry with limited admixture of other bovine ancestors.

## INTRODUCTION

Following the completion of the bovine genome, genome-wide studies have been conducted to catalog genetic variants, e.g., the bovine Hapmap project. Bovine genetic studies have been conducted extensively among European and taurine cattle (*Beja-Pereira et al., 2006*; *Dadi et al., 2014*; *Gautier, Laloe & Moazami-Goudarzi, 2010*; *Gorbach et al., 2010*; *Lee et al., 2014*; *McTavish et al., 2013*; *Speller et al., 2013*; *Suh et al., 2014*). However, little is known about Asian cattle, which are predominantly *Bos indicus*. Recently, a worldwide study of bovines including Asian *Bos indicus* breeds from India, Pakistan, China

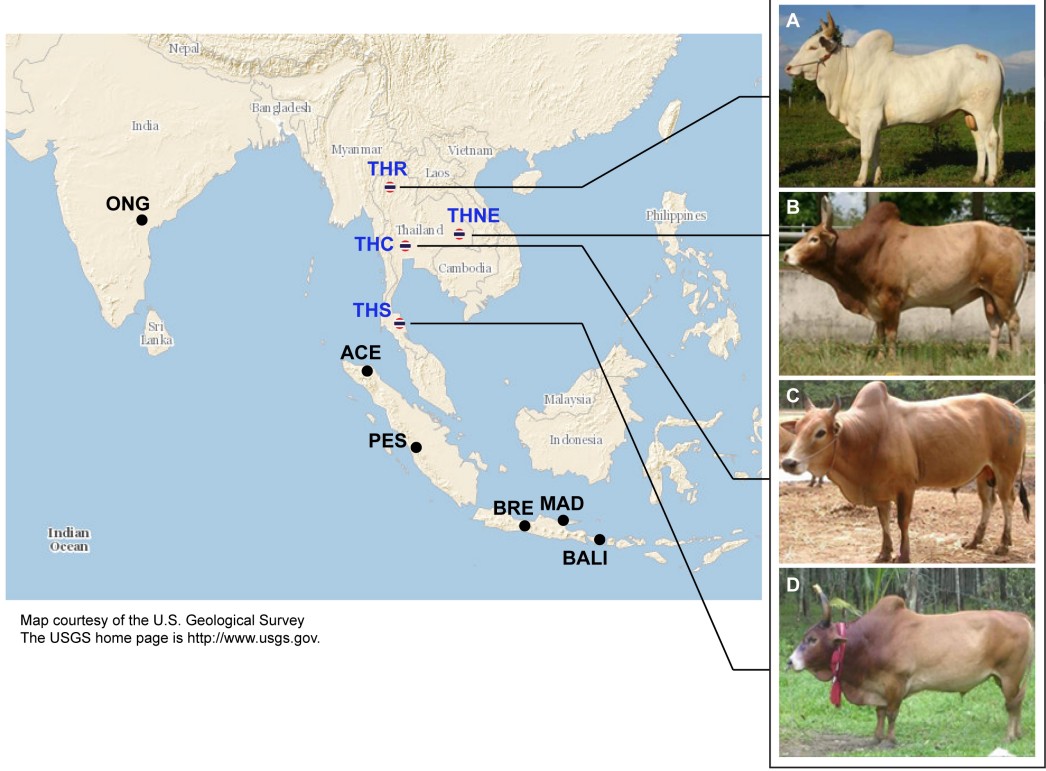

**Figure 1 Images of four native Thai cattle varieties.** (A) Kho-Khaolumpoon (THR), (B) Kho-Isaan (THNE), (C) Kho-Lan (THC), and (D) Kho-Chon (THS) (Image courtesy of *Department of Livestock Development*). The assumed geographical origins of the varieties and other Asian breeds, namely Ongole Grade (ONG), Aceh (ACE), Pesisir (PES), Brebes (BRE), Bali (BALI), and Madura (MAD), are shown.

and Indonesia showed evidence of *Bos javanicus* ancestry that contributes to Asian bovine diversity (*Decker et al., 2014*). However, no one has yet investigated the ancestry of Thai cattle, and the domestication of bovine in Southeast Asia is poorly understood (*Larson & Fuller, 2014*). Therefore, to address this question, further study of Asian *Bos indicus* breeds is needed.

Cattle native to Thailand (excluding recently introduced European breeds) have *Bos indicus* traits, including the distinctive dorsal hump. However, to our knowledge, there has been no formal genetic testing that these cattle are *Bos indicus*. Genetic variation of native Thai bovines has been studied using different marker types, i.e., STR (*Jirajaroenrat, Boonwong & Tuntivisoottikul, 2008a*), mitochondrial DNA (*Jirajaroenrat, Satitmanwiwat & Tuntivisoottikul, 2008b*; *Kornphan et al., 2012*; *Sarataphan et al., 2013*), and SNPs (*Charoensook et al., 2011*; *Edea et al., 2013*). However, none of these studies provide any information of the origins and ancestries of these cattle, since the markers used are insufficiently informative. The earliest evidence of domestic cattle in Thailand dates from 4000 to 5000 B.C.E (*MacHugh, 1996*). There are four native breeds officially recognized by the department of livestock, ministry of agriculture, Thailand, namely Kho-Khaolumpoon (northern Thailand origin, THR), Kho-Isaan (northeastern Thailand, THNE), Kho-Lan (central Thailand, THC) and Kho-Chon (southern Thailand, THS) (Fig. 1). According

to folklore, these breeds may have originated from different regions of Thailand. To our knowledge, there is no historical evidence of any domestication event for these breeds, and so it is unclear if they can be distinguished as breeds in the accepted sense. A breed can be defined as a group of individuals with shared distinctive phenotypic traits and a genetic signature that distinguishes them from other breeds. The phenotypic traits of Thai cattle that are breed characteristics include coat color and body size. THNE coat colors vary from red, black, brown, white to yellow. They have the smallest body size of all Thai breeds. THC and THR have body sizes similar to THNE. The THR breed is considered to possess traits suitable for presentation in the Thai royal plough day ceremony (early May of every year), which may have been performed since the Sukhothai era (13th Century CE). THS coat colors are light/dark brown and black. THS are mainly used as draught animals owing to their large body size relative to other Thai breeds. These Thai cattle are well adapted to the tropical environment, e.g., heat tolerance and resistance to ectoparasites (*Akkahart, 2003*; *Charoensook et al., 2011*; *Kahi, 2004*; *Khamkwan et al., 2012a*; *Khamkwan et al., 2012b*; *Saithong, Chatchawan & Boonyanuwat, 2011*). Notwithstanding these breed characteristics, it is unknown if these four Thai native breeds can be distinguished genetically as breeds, and what their possible origins could be in relation to other cattle.

In this work, we present a population genetics study of 28 individuals sampled from the four Thai native breeds. Genotyping data were obtained using the Illumina BovineSNP50K chip array platform. These data were analyzed together with 1,369 worldwide cattle from 88 breeds previously published (*Decker et al., 2014*).

# MATERIALS AND METHODS

## Ethical statement

In this study, we obtained samples from two sources. Tissue samples were provided by Dr. Yanin Opaspattanakit, Maejo University, Thailand, which were obtained as part of an ongoing Government program, sponsored by the Thailand Research Fund (project code RDC492001) for improving Thai native cattle production (*Department of Livestock Development*. Available from: http://www.dld.go.th/th/images/stories/news/Strategy/55-59%20strategy_beef.pdf). The full list of researchers who contributed the samples in this program is listed in Table S1.

We also obtained blood samples specifically for this study. The studied animals were maintained at the Khon Kaen University (KKU) beef farm, Thailand. This facility is owned by Khon Kaen University for the purpose of Agricultural scientific research. Hence, no field permit is required to study them. Five to ten milliliters of blood were taken from the jugular or tail vein. All efforts were made to minimize distress when taking blood samples. The protocols for recruiting animals and drawing blood were approved by the Animal Ethics Committee of Khon Kaen University under the permit number AEKKU 41/2557.

## Data used

### Thai bovine SNP data

Tissue and blood samples were obtained from 28 individuals. DNA was extracted from tissues using Gu-HCl (*Pramanick, Forstova & Pivec, 1976*). DNA was extracted from
blood using a QIAamp DNA Blood Mini kit (QIAGEN, Hilden, Germany). Final DNA concentrations ranged from 43 to 125 µg/ml (Table S2). Blood samples were obtained from THNE individuals, whereas tissue samples were taken from the rest (THR (8 individuals), THC (5 individuals) and THS (5 individuals)). Genotyping was performed using the Illumina BovineSNP50 chip at the INRA Labogena platform (Jouy-en-Josas, France) using standard procedures (http://www.illumina.com). The raw genotypic data are available in Dataset S1. The same data quality control process as described in *Gautier, Laloe & Moazami-Goudarzi (2010)* was used, which left 44,706 markers passing the quality control. These markers were used in all subsequent analyses. To check whether closely related individuals were sampled, we performed identity-by-decent (IBD) test as recommended by *Porto-Neto et al. (2013)*. IBD values were calculated in pairwise combinations of individuals using PLINK software (*Purcell et al., 2007*).

***Worldwide bovine SNP data***

The dataset containing 1,365 individuals from 87 worldwide bovine breeds was downloaded from http://dx.doi.org/10.5061/dryad.th092, published in *Decker et al. (2014)*. This dataset includes data from other studies (*Gautier et al., 2009*; *Gautier, Laloe & Moazami-Goudarzi, 2010*). We also included four individuals of *Bison bison* (OBB) described in *Gautier, Laloe & Moazami-Goudarzi (2010)*. Full information of sampled individuals is shown in Table S3.

## Fst and informative marker selection genetic distance among bovine breeds

In this analysis, the Thai individuals were assumed to belong to the same group of Thai cattle. All other cattle were grouped according to breed labels. Genetic distance between breeds according to groups of individuals was calculated using the pairwise Weir & Cockerham Fst equation (*Weir & Cockerham, 1984*).

## Phylogenetic analysis

Bootstrapped phylogenetic trees were constructed using Phylip (*Felsenstein, 2005*). A pairwise distance matrix among breeds was constructed using Nei's genetic distance. All Thai cattle were put into one group. One hundred bootstraps were performed with random 10% marker resampling. The consensus tree was constructed from the bootstrapped data with bootstrapping confidence values assigned to each node using FigTree version 1.4.2 (graphical viewer accompanying BEAST; *Drummond & Rambaut, 2007*).

## Unsupervised population clustering

Iterative Pruning Principal Component Analysis (ipPCA; *Limpiti et al., 2011a*; *Limpiti et al., 2011b*) was used to cluster individuals into subpopulation groups. This unsupervised clustering algorithm uses Principal Component Analysis (PCA). The EigenDev heuristic is employed on the Eigenvalues from PCA to detect whether substructure exists. If this condition is met (EigenDev $>0.21$), the individuals are separated into two groups using high-dimensional fuzzy c-means clustering. The procedure is iterated until no

substructure exists among the individuals, which are defined as subpopulations. The ipPCA algorithm can be used to assign individuals to groups without assumptions of ancestry from population, i.e., breed labels. The theoretical basis for why the iterative clustering approach used in ipPCA can provide greater accuracy is described in *Limpiti et al. (2011a)* and *Limpiti et al. (2011b)*.

We extended the ipPCA analysis framework by testing the robustness of subpopulation assignments. In this extra procedure, 10% of the markers were randomly sampled (with replacement) using a random generator library implemented in MATLAB version R2009b, and the resampled genotypic data were used as input for ipPCA. One thousand bootstrap-resampled datasets were generated and the consensus subpopulation assignment map was constructed. The bootstrap confidence values were calculated from the percentage of individuals assigned in all bootstrap datasets (Table S4).

### ADMIXTURE analysis

The ADMIXTURE software (*Alexander, Novembre & Lange, 2009*) was used for inferring ancestry ratios. This program uses maximum likelihood modeling to estimate ancestry, which is much faster than Bayesian modeled ancestry as implemented in STRUCTURE (*Pritchard, Stephens & Donnelly, 2000*). The number of $K$-ancestors was varied from $K = 2$ to $K = 100$ (whole dataset), and $K = 2$ to $K = 30$ (indicine only). Ten cross-validations were performed to estimate cross-validation error for determining the suitable number of $K$-ancestors for interpretation. The cross-validation plots are shown in Fig. S1. Individual ancestry patterns were generated using CLUMPP (*Jakobsson & Rosenberg, 2007*) from the ADMIXTURE outputs. Individuals in Fig. 4A were sorted according to the subpopulation assignments made by ipPCA (see Dataset S2 for the full result of the ADMIXTURE analysis). To test whether cryptic relatedness among Thai individuals could cause artifactual groupings of inferred ancestry in ADMIXTURE analysis, ADMIXTURE was performed with 268 indicine individuals plus a Thai individual. This process was repeated for all 28 Thai individuals. The $Q$ value of the Southeast Asian ancestral component, which is the major ancestral component value among PES and ACE Indonesian breeds, was extracted from the Thai individual in each analysis.

## RESULTS AND DISCUSSION

We used four different approaches to ascertain the population structure and genetic identities of Thai native cattle. First, pairwise comparisons of all breeds were performed using Fst analysis. In this analysis, the Thai individuals were placed into one group as we have no prior evidence that the four breeds are actually genetically distinct groups. The Thai cattle are clearly identified as *Bos indicus* (Fig. 2). Next, to discern in more detail the relationship of Thai to other indicine cattle, NJ tree analysis was performed. Thai cattle occupy a position in the tree among other Southeast Asian indicine breeds (PES, ACE, BRE, MAD, and BALI) with 100% bootstrap support (Fig. 3). In particular, PES and ACE breeds are on a node closer to Thai than BRE, MAD and BALI. The BRE and MAD breeds are *Bos indicus* with considerable *Bos javanicus* ancestry (represented by BALI) (*Decker et al., 2014*). Hence, Thai cattle are less likely to have admixture of *Bos javanicus*. Although

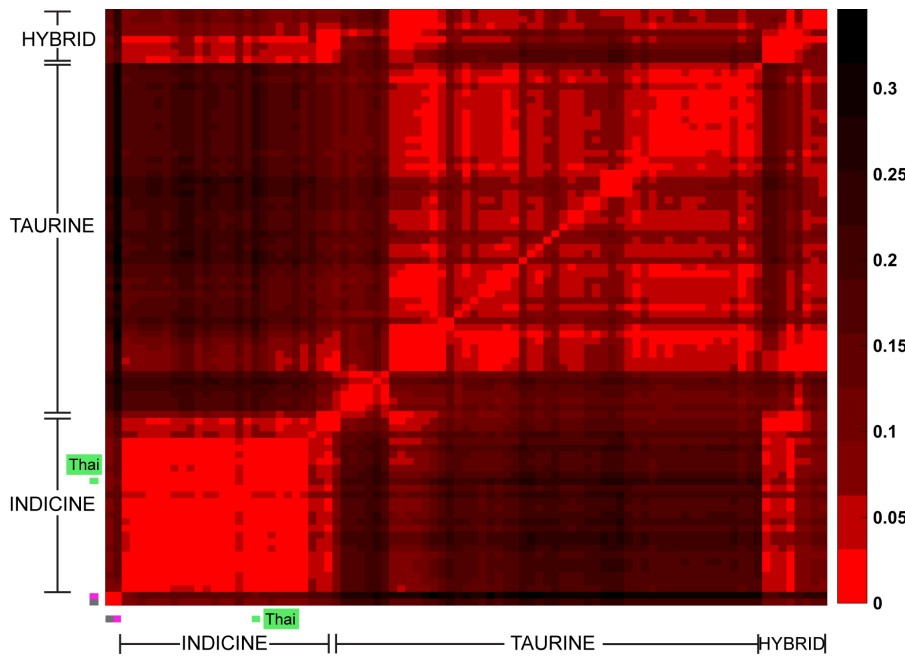

**Figure 2 Heat map of pair-wise Fst.** Each cell represents a group of individuals of the same breed; the heat map colors reflect Weir & Cockerham Fst values as shown by the color scale bar on the right. The data from 1,397 animals genotyped for 44,706 markers were used to calculate Fst values. The gray cell represents Bison bison (OBB) and the magenta cell represents Bos javanicus (BALI), which are non-cattle reference animals. The Thai native cattle (28 individuals from THS, THR, THC, and THNE) are indicated by a green bar. Indicus, Taurine, Indicus/Taurine hybrid breeds are grouped arbitrarily, as indicated by brackets.

the Thai cattle occupy a distinct position in the tree with their Southeast Asian neighbors, there is ambiguity in the relationship to other Asian indicine breeds (RSIN, DHA, KAN, and SAHW) as shown by the nodes with weaker support (<80%).

We tested the hypothesis that cattle breeds define genetically homogenous subpopulations of individuals using a clustering approach that is blind to the breed information. This approach, implemented using the ipPCA algorithm, identified 96 genetically distinct subpopulations (Table S5). The majority of these subpopulations have strong bootstrap supports in ipPCA, which in the main comprise individuals with the same breed label. However, a direct correspondence between breeds and genetically homogenous subpopulations does not always exist. For example, SP28, SP29 and SP30 comprise individuals from the same NORM breed (see Fig. S2). Moreover, some confidently assigned (≥95% bootstrap) subpopulations comprise individuals with different breed labels, including SP4-9, SP11-13, SP15, SP18-33, SP35-40, SP42-60, SP62-64, SP68-76, SP79-81, SP89, SP90, SP93, SP95, and SP96. Genetically distinct subpopulations among individuals of the same breed may arise from drift, particularly for breeds that were established a long time ago. Conversely, breeds with recent common ancestry may not be distinguished as genetically distinct subpopulations with the genetic markers available. Although ipPCA assignments were robust in the main, some subpopulations had weak bootstrap support (≤80%) indicating that assignments of these individuals are inaccurate.

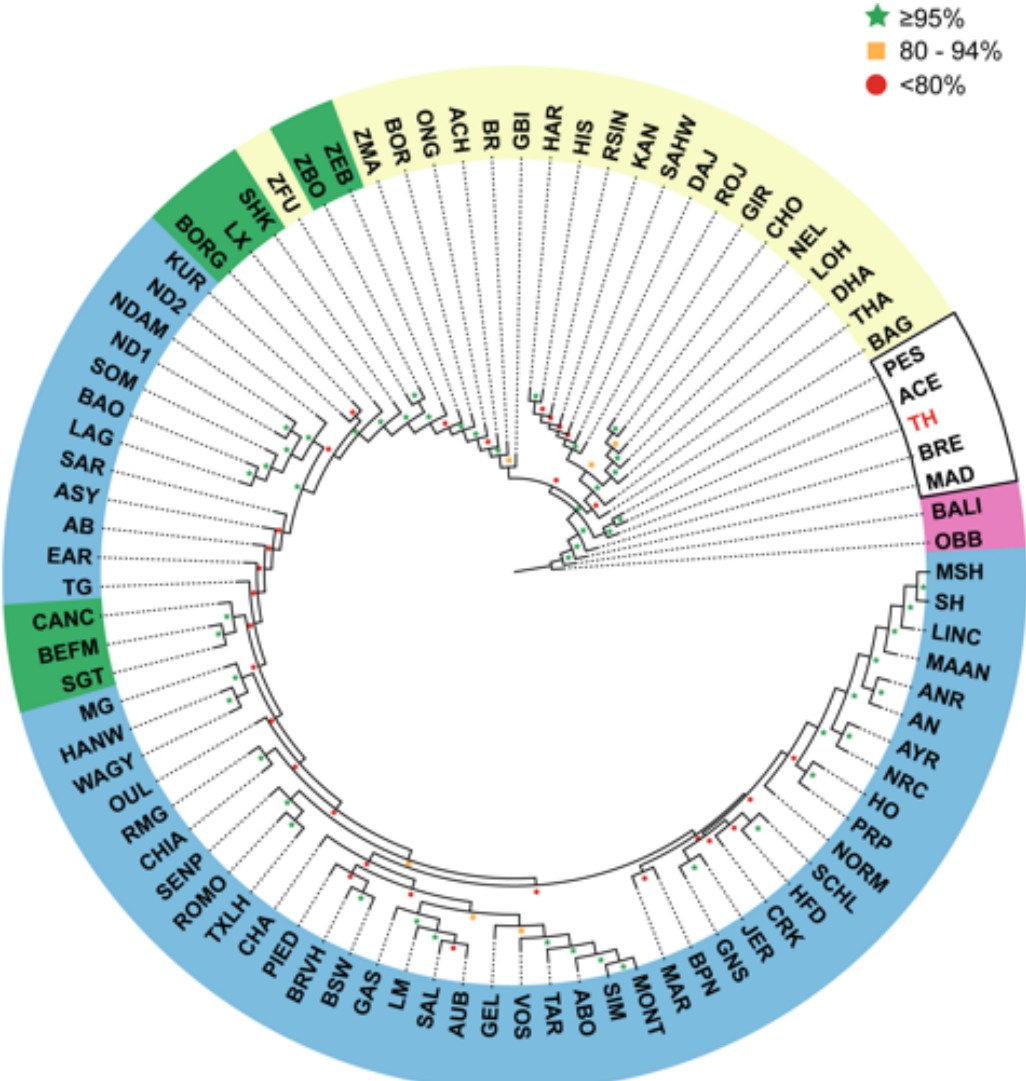

**Figure 3 Unrooted Neighbor-Joining phylogenetic tree of cattle breeds.** A pair-wise distance matrix between breeds was calculated based on minor allele frequency (MAF). The data from 1,397 animals genotyped for 44,706 markers were used to calculate MAF values. The consensus tree is shown from 100 bootstrap replicates. The bootstrap confidence intervals (≤80%, red circle; 80–95%, yellow square; ≥95%, green star) are indicated at each node. Taurine breeds are highlighted in blue, Hybrid breeds in green and Indicine breeds in yellow. Southeast Asian cattle are outlined in black. The Thai cattle (TH, 28 individuals from THS, THR, THC, and THNE) are highlighted in red. The non-cattle out-groups Bison bison (OBB) and Bos javanicus (BALI) are highlighted in pink.

The inaccuracy in cluster assignment by ipPCA for these subpopulations reflects the weak bootstrap support of the same individuals in the NJ tree, e.g., SP85-87 contain individuals from South Asian breeds (THA, RSIN, HAR, KAN, CHO, DAJ, LOH, and ACH). Cluster assignment of Southeast Asian breeds from Indonesia is also somewhat inaccurate in that only ACE (SP91) is likely to be a genetically distinct group (88% bootstrap support).

Thai cattle were assigned to SP88, SP89, and SP90. THS and THNE individuals comprise SP89 and SP90 respectively, which were assigned with strong bootstrap support

Wangkumhang et al. (2015), *PeerJ*, DOI 10.7717/peerj.1318

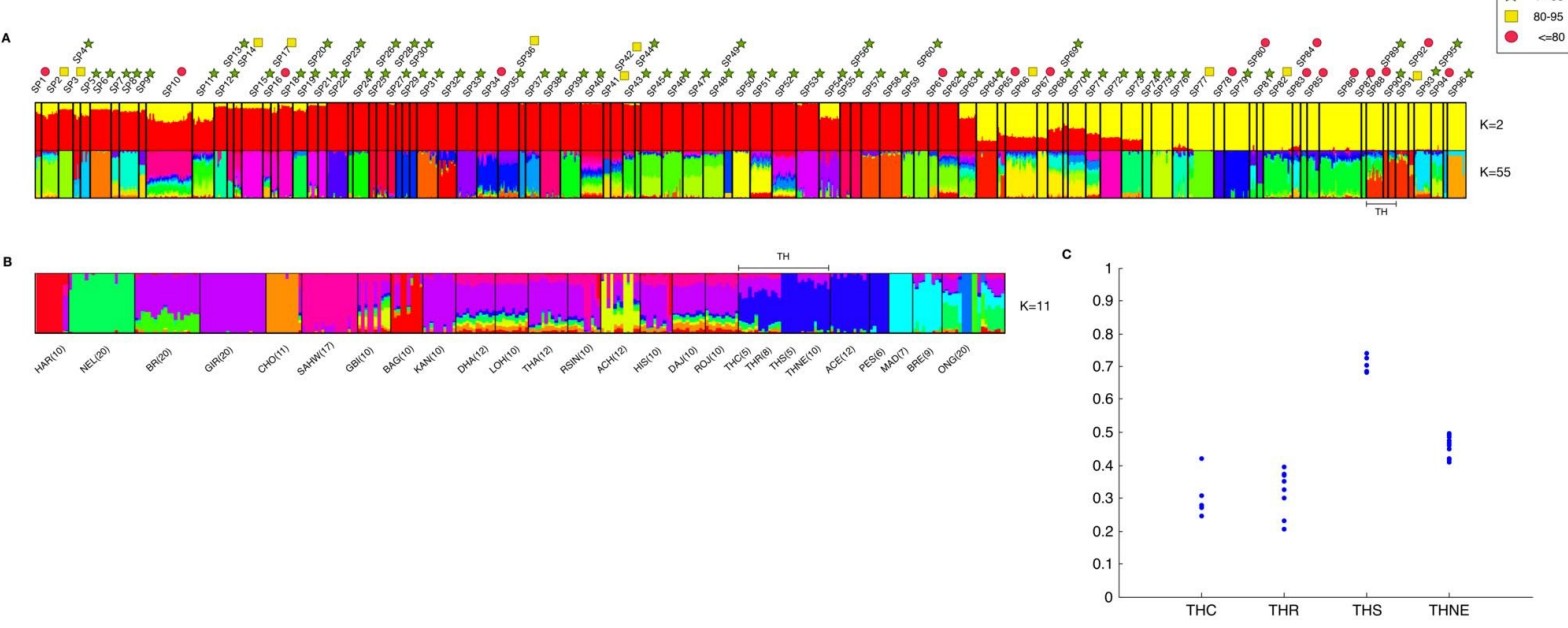

**Figure 4 Combined result of ipPCA and ADMIXTURE analysis.** The data from 1,397 animals genotyped for 44,706 markers were used in ipPCA to assign individuals to subpopulation groups (SP1–SP96). Each SP contains individuals with no significant substructures detected by ipPCA. The SP assignment is independent of the breed label information. To assess the confidence of SP assignment of ipPCA, bootstrapping was performed. Admixture analysis was then performed on the same data with cross-validation to determine optimal $K$. The number of $K$ ancestral components was varied from $K = 2$ to 100. (A) ADMIXTURE result from $K = 2$ and $K = 55$. The latter $K$ is the minimum cross-validation error reported by ADMIXTURE. The complete results of all $K$s are shown in Dataset S2. SP74-94 contain indicine breed individuals only, and the 28 Thai individuals are assigned to SP88-90 (bracketed by TH). The ADMIXTURE plots show the proportion of each inferred ancestral component, as indicated by different colors for each individual. The individuals were sorted horizontally according to the 96 subpopulation assignments made by ipPCA. Bootstrap intervals ($\leq$80%, red circle; 80–95%, yellow square; $\geq$95%, green star) for each ipPCA assigned SP are indicated by the symbols on the top. The plots for $K = 2$ and 55 are shown; (B) ADMIXTURE analysis of indicine individuals only (296 individuals genotyped for 44,706 markers) was performed varying ancestral components from $K = 2$ to 30. The plot shows the result from $K = 11$, which is the $K$ with minimum cross-validation error. The individuals are sorted according to breed labels shown underneath the plot; the 28 Thai individuals are grouped according to their assumed origins (THC, THR, THS, and THNE). (C) Ratio of inferred Southeast Asian ancestral component among Thai cattle. 28 ADMIXTURE analyses were performed by introducing a single Thai individual to 268 indicine cattle from other non-Thai breeds. The minimum cross-validation error was observed at $K = 8$ in all 28 ADMIXTURE runs. The plot shows $Q$ values of the inferred major Southeast Asian ancestral component among the 28 Thai individuals grouped according to their assumed origins (THC, THR, THS, and THNE).

(≥95%). In contrast, SP88 comprises individuals from three Thai breeds, THC, THNE, and THR with weaker bootstrap support. Therefore, three distinct subpopulations can be resolved among the Thai cattle. It is interesting to note that THR, which is revered as a Thai cultural icon, is genetically similar to other Thai cattle (THNE and THC). The only Thai breed that appears as genetically distinct group is THS.

Ancestry was determined by ADMIXTURE analysis. At $K = 2$ modeled ancestors, the major ancestral component of taurine cattle is clearly distinct from the major component of indicine cattle (Fig. 4A). Taurine cattle were assigned predominantly to SP12–SP53 by ipPCA, and they share a major ancestral component (red). In contrast, ipPCA subpopulations SP64–SP94 comprise indicine individuals that share the other major ancestral component (yellow). Individuals of hybrid breeds, e.g., Beefmaster (BEFM) assigned to SP54, exhibit both ancestral components. According to the cross-validation calculation in ADMIXTURE, $K = 55$ has the lowest cross-validation error and thus it is a suitable number of ancestral components for the entire population dataset (Fig. S1A). Patterns specific to each subpopulation are apparent at $K = 55$, in particular the groups with assigned European taurine individuals. Within the subpopulations containing mostly indicine individuals, the individuals within SP88–SP94 share a unique major ancestral component.

Having shown that Thai cattle are indicine, we investigated ancestry patterns within the indicine cattle in greater detail by performing ADMIXTURE on the indicine cattle only. According to the cross-validation calculation in ADMIXTURE, $K = 11$ is a suitable number of ancestral components for the indicine sub-dataset (Fig. S1B). The Southeast Asian ACE, PES and Thai cattle shared a major ancestral component (dark blue) distinct from other indicine, including MAD and BRE Southeast Asian cattle (Fig. 4B). This result is in agreement with the NJ tree shown in Fig. 2, in which MAD and BRE are distinct from other Southeast Asian cattle because of *Bos javanicus* admixture. Among the Thai cattle, the THS have the least admixture of other components. The distinctiveness of the THS individuals could be the result of cryptic relatedness, since IBD scores are markedly higher than 0.1 (Table S5). Sampling of relatives is not congruent with the ADMIXTURE model assumption that individuals are independent (*Alexander, Novembre & Lange, 2009*). To reduce the possible effect of cryptic relatedness among Thai individuals, we performed another set of ADMIXTURE analyses in which indicine cattle were analyzed with a single Thai individual, and repeated for all 28 Thai cattle. In these analyses, the optimal $K$ inferred from cross-validation error was $K = 8$ for all 28 experiments of ADMIXTURE (Fig. S1C). From the NJ analysis and earlier ADMIXTURE analyses, we used the assumption that the major ancestral component of ACE and PES individuals was shared with the Thai cattle. The variation of this Southeast Asian ancestry among the Thai individuals in shown in Fig. 4C. The THS individuals have a markedly greater proportion (0.65–0.77) of Southeast Asian ancestry from the others (THC, THR, and THNE), such that THS can be considered as a genetically distinct breed. It should be noted that the sampling of THS and other Thai cattle is rather low and could be biased. Therefore, the conclusion about the uniqueness of these cattle must take this caveat into account. The uniqueness of the Southeast Asian cattle is perhaps somewhat surprising given the

known ascertainment bias of SNP markers toward taurine breeds (*Matukumalli et al., 2009*; *McTavish et al., 2013*). This is obvious from our admixture analysis in which the *Bison bison* and *Bos javanicus* are indistinguishable from indicine breeds at low numbers of $K$ ancestors ($K = 2$–$5$; Dataset S2).

In conclusion, Thai cattle are indicine that are most closely related to other Southeast Asian breeds. The Kho-Chon variety (THS) appears to be a distinct breed with minimal admixture. All other Thai and Southeast Asian cattle show evidence of admixture. However, this claim must be tempered against the limitation of the data available, in particular the marker platform and sampling of individuals. Moreover, the weak bootstrap support from NJ tree and ipPCA clustering among South Asian indicine breeds further points to lack of power to differentiate indicine breeds. Future studies into bovine genetic diversity should include whole genome sequencing to discover new variants among indicine breeds. Kho-Chon Thai cattle are of particular interest for further genomic study into the Southeast Asian *Bos indicus* ancestor.

## ACKNOWLEDGEMENT

We thank Dr. Yanin Opaspattanakit, Maejo University for providing tissue samples. We also acknowledge Ms. Thongsa Buasook and Dr. Thevin Vongpralab, Department of Animal Science, Faculty of Agriculture, KhonKaen University for their assistance in collecting blood samples. We acknowledge the Thai Department of Livestock Development, Ministry of Agriculture and Cooperatives for providing images of the Thai native cattle. The geographical map used in Fig. 1 is courtesy of the US Geological Survey. Finally, we thank the Editor and reviewers for their suggestions to improve the manuscript.

### Funding

ST is supported in part by the Thailand Research Fund (TRF grant number RSA5880061) and the National Science and Technology Development Agency, 2011 Research Chair Grant. PJS is supported by The Thailand Research Fund (TRF grant number RSA5780007). The funders had no role in study design, data collection and analysis, decision to publish, or preparation of the manuscript.

### Grant Disclosures

The following grant information was disclosed by the authors:
Thailand Research Fund: RSA5880061, RSA5780007.
National Science and Technology Development Agency.

### Competing Interests

The authors declare there are no competing interests.

### Author Contributions

- Pongsakorn Wangkumhang analyzed the data, wrote the paper, prepared figures and/or tables, reviewed drafts of the paper.

- Alisa Wilantho, Katayoun Moazami-Goudarzi and Mathieu Gautier performed the experiments, analyzed the data, prepared figures and/or tables.
- Philip J. Shaw and Sissades Tongsima conceived and designed the experiments, analyzed the data, wrote the paper, reviewed drafts of the paper.
- Laurence Flori performed the experiments, analyzed the data, prepared figures and/or tables, reviewed drafts of the paper.
- Monchai Duangjinda performed the experiments.
- Anunchai Assawamakin analyzed the data.

### Animal Ethics

The following information was supplied relating to ethical approvals (i.e., approving body and any reference numbers):

In this study, we obtained samples from two sources. Tissue samples were provided by Dr. Yanin Opaspattanakit, Maejo University, Thailand, which were obtained as part of an ongoing Government program, sponsored by the Thailand Research Fund (project code RDC492001) for improving Thai native cattle production (*Department of Livestock Development*. Available from: http://www.dld.go.th/th/images/stories/news/Strategy/55-59%20strategy_beef.pdf). The full list of researchers who contributed the samples in this program are listed in Table S1.

We also obtained blood samples specifically for this study. The studied animals were maintained at the Khon Kaen University (KKU) beef farm, Thailand. This facility is owned by Khon Kaen University for the purpose of Agricultural scientific research. Hence, no field permit is required to study them. Five to ten milliliters of blood was taken from the jugular or tail vein. All efforts were made to minimize distress when taking blood samples. The protocols for recruiting animals and drawing blood were approved by the Animal Ethics Committee of Khon Kaen University under the permit number AEKKU 41/2557.

### Supplemental Information

Supplemental information for this article can be found online at http://dx.doi.org/10.7717/peerj.1318#supplemental-information.

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
