# Peer review of "Genetic analysis of Thai cattle reveals a Southeast Asian indicine ancestry"

_PeerJ, doi:10.7717/peerj.1318_

## Round 0.1 · original submission · Major Revisions

· Academic Editor

Major Revisions

This paper has now been seen by two reviewers and I have read it myself. One referee suggests Reject and another Accept with minor corrections. As usual in such cases the fairest decision tends to be in between.

The strongest criticism is based on sample size and I agree this is a big issue. I agree entirely that analyses based on STRUCTURE (and its derivatives) and Fst become less and less reliable as sample size goes down. This is particularly true when, as here, there is a strong danger that relatives will be sampled. There seem to be two main options for rescuing the data and making them publishable:

1. Ideally the data are extended to include more individuals. Regardless, it is important to include more details about sampling so that the reader can feel confident that results are not distorted by inclusion of relatives. If relatives are (unavoidably) included this needs to be made clear.

2. Although not perfect, the issues of small sample size and possible inclusion of relatives can reasonably be addressed by running ADMIXTURE once for each individual in the Thai group, each time including a single Thai individual. This is a form of cross-validation, the idea being that one sample has negligible impact on resolution of the K-groups so each Thai cow will obtain assignment values for each group. The emerging picture will be one where the authors can assess how strongly the Thai cows fit to each of the K-groups and the extent of variation between individuals. I believe this would achieve the key aim of the paper.

While the above options would be acceptable, to me only an increase in sample size would make the relationships among the Thai breeds (Fig 5) acceptable and I suggest this section is dropped until larger sample size can be obtained. The problem is that the changes in log likelihood are only trustworthy if a whole range of underlying assumptions is met (independence of mutations, lack of genotyping error, lack of relatedness between individuals, homogeneous population structure etc. etc.). Since many or most of these are not met, the small and unstable differences in node order should be viewed as within experimental error.

The Referees make a number of other important points, particularly the requests for more information / discussion as to why previous related studies failed / cannot be trusted and elements of the historical background. I also think that more explanation is needed about the methods. For example, making it clear that ADMIXTURE is effectively STRUCTURE implemented with a faster algorithm and that it finds groups rather than “ancestors” as stated in the current draft. I was also left a little confused by the choice of bootstrap ranges for Fig. 3. A threshold of 56 seems arbitrary. I would prefer 95%-100% (= solid), 80%-94% (= reasonably solid) and less than 80% (= not reliable). With so many data, nodes that are not supported close to 100% should be considered dubious.

In summary, the paper needs Major Revisions to become acceptable. I do tend to agree with the more negative Referee that sample sizes need to be increased. However, I can equally accept that this may be logistically challenging or outside the scope of current funding. Consequently, I prefer to leave open a door based on substantial reworking of the key analyses. The authors need to expand their text somewhat to provide fuller explanations in a number of places. Reviewer 2 makes a number of useful suggestions that should be considered.

·

Basic reporting

This is a potentially interesting study of the origins of Southeast Asian cattle. The authors have analysed 28 cattle from Thailand from 4 breeds using the Bovine 50K SNP array. The data were analysed in several relatively standard ways and combined with 1,365 individuals from 87 breeds.

One of the key criteria of a population genomics paper is that it represents substantial data to answer the questions posed. In this particular case there are n = 5, 5, 8, and 10 animals in the 4 breeds for which data were collected. Some of the analyses were based on allele frequency data, and obviously such small samples will not give accurate allele frequency estimates. Furthermore, in some cases Fst values were calculated and it is well known that Fst values do not start to converge until at least 10 animals are used. Population admixture analyses were performed, some of which use an underlying departure from Hardy-Weinberg Equilibrium, and again, with so few samples this would not give accurate estimates.

Examining the current literature one can see many examples of analyses using these sorts of methods and invariably they have much larger sample sizes.

I note that more animals were initially sampled but nearly half the sample of animals was of DNA that was too degraded to use. Nevertheless, the sample is too small to be sure that the results would be relatively unchanged with the addition of more animals, and so one would recommend the addition of more unrelated animals from the 4 Thai breeds.

Signed:

William Barendse

Experimental design

No further comments

Validity of the findings

No further comments

Additional comments

no further comments

Reviewer 2 ·

Basic reporting

This study presents a novel data set and analysis of Thai cattle. The conclusions are quite robust and interesting. The authors conclusively show that Thai cattle are indicines (as expected although, apparently, never actually shown before) and show some interesting structure patterns within Thailand (i.e. relationship between Southern and Indonesia breeds). The paper well written besides few specific problems (see my specific comments below).

However, I have raised a few major (and minor) issues that I think needs to be addressed before publication. In particular I think the author rely too much on the population history modelling which seems to give some very unexpected results (i.e. population sizes). I have suggested a few extra analyses that could be easier to interpret and yield more robust results.
There are also a few statements that need clarification. In particular statements in which the author argue for a domestication centre in Thailand (see specifics). I would recommend to the author this very good review on the concept of domestication: Larson and Fuller (2014). The Evolution of Animal Domestication. Annual Review of Ecology and Evolution.

Specific comments
Major:

Ln 59-60: “The earliest evidence of bovine domestication in Thailand dates from 4000–5000 B.C.E (MacHugh, 1996).” This sentence needs more evidence than just MacHugh’s thesis. Do the authors imply that a separate cattle domestication process took place in Thailand or that this date is the earliest evidence of domestic cattle in Thailand? These are very different statements. The former seems unlikely to me. The latter imply that Thai cattle were imported by humans rather than locally domesticated.

The ADMIXTURE analysis lacks cross-validation procedures. This is necessary, as it provides the reader with the necessary information to assess the fit of the different models (K values).

Ln 222-5: “The topology with maximum likelihood (Fig. 5A) shows that THS lies along the longest branch and is directly 
connected to the ancestral root of all four breeds. We infer that THS is the closest to the ancestor of all Thai cattle. ” I am not sure I understand this statement. Firstly, is that a rooted tree? How was it rooted? Secondly, these two sentences are contradictory. We are told that the branch to the THS is the longest, yet it is the closest to the root? I think the author means that THS branch off first but this needs clarifications.
The authors also suggest that in Figure 5B-F THS is further away from the root. I am not convinced that the distance from THS to the root of the node is longer in 5B than 5A. Please also note that many of these figures are equivalent unrooted trees.
Ln 228-9: “Current population estimates of Thai breeds are THS, 450,000; THNE, 1,230,000; THC, 400,000; and THR, 350,000 “ Is that Ne? That seems huge for a large mammal.

I suggest that the author should run an ADMIXUTURE analysis solely with the Thai breeds and compute the best K value (using cross validation). This will provide the reader with a better overview of the substructure between these breeds but also inform us about which breed are more closely related. For example, if the authors are right about the THS breed as being distinct, it should be obvious at K2.

ADMIXTURE could also be used to assess “purity” among Thai breeds. For example, the authors could select a few indicines populations and see which of the Thai breeds is the most specific (less “admixed”).

Minor:

This sentence in the abstract: “However 
very little is known of the history of how these cattle were domesticated and so questions remain 
of their origins and relationships to other breeds. “ Feels a bit off. Do you imply that there was a separate domestication process in Thailand or do you mean we know little about the domestication of indicines?

Ln 49-50: “While these data are important contributions to our understanding of bovine domestication, we cannot conclude that all Asian Bos indicus breeds share similar patterns of ancestry. “ Same here, this sentence feels off. Why not just say something like: “However, no one has yet investigated the ancestry of Thai cattle.

Ln 59: “However, these studies provide insufficient genetic information to reveal the origins and ancestries of these cattle. ” How come SNPs, STR and mtDNA studies have not been able to conclude whether Thai cattle are indicines or not? Sample size? Lack of analysis? Please explain.

Figure 2: I very much like this heatmap representation. A box around the Thai cattle in the heatmap would ensure they stand out.

Ln 143: I am not sure this is really bootstrapping. Did you perform a re-sampling with replacement?

Experimental design

NA

Validity of the findings

NA

Additional comments

NA

---

## Round 0.2 · Major Revisions

· Academic Editor

Major Revisions

A number of the original criticisms have been addressed. However, for me, the critical one remains. An IBD of 0.8 is far, far too high to be considered independent in a STRUCTURE-type analysis. Although I have not conducted formal tests, my intuition is that anything above 0.1 or even 0.05 will be problematic. After all, STRUCTURE is effectively using the null hypothesis that IBD is zero.

As I said in my original decision letter, the best way around this would be to analyse each Thai sample separately so that the high IBD has no effect. In this way the small sample and high relatedness of the Thai cattle cannot act to create their own groups. Instead, each Thai individual will be placed in one of the other groups suggested by Admixture and consistent placement will be indicative both of genetic affinity and purity of signal (though some caution needs to be used because their relatedness will tend to cause them to be placed in together; they are not independent obsevations).

The statement that Porto-Neto et al. recommend an IBD value of <0.8 is badly misleading. Although these authors use this threshold they do not conduct the sort of analysis that would be required to make any kind of general recommendation for other studies. Instead, they conduct a radically different type of analysis in which K is fixed at K=2 and look to see the fidelity of assignment of cattle from many breeds to the two classes. Here, IBD is much less important because each breed contributes negligibly to the nature of the two groups, so the very high threshold is defendable. In the current study K is varied to find the best-fit number of groups. Here, any appreciable IBD will tend to create unreliable groups due to the IBD.

In view of the above the paper cannot be accepted in its current form because the Admixture analysis cannot be deemed reliable. The authors may decide to follow my suggestion or to find their own solution but they must make a solid case that IBD has been controlled sufficiently for the current application of Admixture.

Finally, the current version is also a little sloppy. There are quite a number of minor typographical and grammatical errors that are picked up by standard error checks. Also the Supplementary Figures (in particular) appear to have been put together without proper attention to detail: most importantly, the legends do not contain enough information for a reader to understand what the figures show and there are formatting issues such as failure to format numbers sensibly (e.g. 0.35 is pointlessly shown as 0.3500).

---

## Round 0.3 · accepted · Accept

· Academic Editor

Accept

I am pleased to see the new analysis which I find convincing and am now happy to recommend acceptance of the manuscript.